# Optimal Substrate Moisture Content for Kiwifruit (*Actinidia valvata* Dunn) Seedling Growth Based on Analyses of Biomass, Antioxidant Defense, and Photosynthetic Response

Dan-Dan Peng [1,†], Da-Gang Chen [1,†], Kai-Wei Xu [1,*], Petri Penttinen [1], Hao-Yu You [1], Hui-Ping Liao [2], Ran Yang [1] and Yuan-Xue Chen [1,*]

[1] College of Resource, Sichuan Agricultural University, Chengdu 611130, China; 71343@sicau.edu.cn (D.-D.P.); 15738269329@163.com (D.-G.C.); petri.penttinen@helsinki.fi (P.P.); youhaoyu0922@163.com (H.-Y.Y.); 18227829629@163.com (R.Y.)
[2] Sichuan Huasheng Agricultural Co., Ltd., Deyang 618200, China; schsny@163.com
[*] Correspondence: xkwei@126.com (K.-W.X.); cyxue11889@163.com (Y.-X.C.)
[†] These authors contributed equally to this work.

**Abstract:** The fruits of kiwifruit are well known for their abundant nutritional value and health benefits, but kiwifruit vines are susceptible to environmental factors such as drought or waterlogging. Optimum substrate moisture content (SMC) can decrease cultivation costs and improve the quality of seedlings in soilless cultivation. To quantify the water requirements of kiwifruit seedlings, a greenhouse study was conducted to investigate the growth, antioxidant defense, and photosynthetic parameters of seedlings of *Actinidia valvata* Dunn at six levels of SMC (20%, 40%, 60%, 80%, 100%, and 120%). Results showed that shoot and root dry matter accumulation increased gradually with the increase in SMC from 20% to 100% and was lower at 120% SMC than at 100% SMC. Electrolyte leakage and malondialdehyde content were the lowest at 80% and 100% SMC. Antioxidant enzyme activities, including superoxide dismutase, peroxidase, and catalase, chlorophyll content, net photosynthetic rate, maximal quantum yield of PSII photochemistry, photosynthetic electron transfer rate, and actual quantum yield were the highest at 80% and 100% SMC, but there was no significant difference in these parameters between the two treatments (80% and 100% SMC). However, the shoot and root dry weights of seedlings at 100% SMC were 13.20% and 33.02% higher than those at 80% SMC, respectively. In summary, 100% SMC provided optimal water supply for the photosynthetic efficiency and dry matter accumulation of shoots and roots. The results are expected to be useful for the mass production of high-quality kiwifruit seedlings in greenhouse or nursery containers, with the potential to save water.

**Keywords:** substrate moisture content; water requirements; growth; plant physiology; water stress

## 1. Introduction

Kiwifruit (*Actinidia* sp.) is a perennial deciduous vine and its fruits are popular worldwide owing to their unique flavor and high nutritional value, especially for their high vitamin C content [1,2]. Kiwifruit are cultivated all over China, and the orchard area of 184,554 ha and fruit production of more than 2.23 million tons by 2020 were the highest in the world [3]. However, the yield per unit area lags other countries because of poor cultivation and management technologies [4]. Soilless cultivation is considered as a priority alternative for rapid propagation of high-class kiwifruit seedlings. The main advantages of soilless culture are better control of soil-borne diseases, fertilization, and irrigation, which improves the yield and quality of seedlings [5,6]. Currently, soilless substrate has been successfully used for cultivation of various fruit tree seedlings, including blueberries (*Vaccinium*. sp. L.) [7–9] and fig trees (*Ficus carica* L.), with better growth compared to traditional cultivation in soil [10,11]. Our previous studies also showed that the growth

and quality of kiwifruit seedlings cultivated in soilless substrate were higher than those cultivated in soil. An appropriate water supply is essential for the growth and quality of seedlings in a soilless cultivation system. Efficient irrigation practice improves the water use efficiency of plants and reduces leaching and runoff of water and nutrients, thereby decreasing cultivation costs [12]. In addition, an appropriate water content in soilless culture systems could effectively reduce the occurrence and development of root diseases [13]. However, knowledge on the optimal range of substrate moisture content (SMC) for soilless cultivation is limited, and the range varies with different soilless culture substrates and species.

Water availability around the root system directly affects the physiological responses of above-ground parts. Estimating crop water requirements by measuring the physiological responses of plants has been proposed as an effective way to minimize or prevent over-irrigation and leaching [14–16]. Photosynthesis is closely related to stomatal conductance, and both are extremely sensitive to water restriction [14,17–19]. Irrigation schedules based on the relationship between photosynthesis and SMC in container nurseries have received considerable attention [12,15,20–23]. For example, when the irrigation requirements of landscape roses were determined according to growth and photosynthetic responses to SMC, the shoot and root dry matter, net photosynthetic rate ($P_n$), stomatal conductance ($g_s$), and transpiration rate ($T_r$) were highest at 30% to 40% SMC [12]. Similar results were found in cultivating *Crepidiastrum denticulatum* at 20% to 60% SMC, which showed that the photosynthetic and antioxidant capacity first increased and then decreased, and the plants grew best and the antioxidant capacity was highest at 45% SMC [20]. In addition, water stress disrupts the balance of the cellular redox state, resulting in the accumulation of reactive oxygen species (ROS), which can cause severe damage by oxidizing membrane lipids and the photosynthetic apparatus [24,25]. Chlorophyll (Chl) fluorescence indicating the photochemical efficiency of photosystem II (PSII) has been widely applied to assess water stress-induced damage to the photosynthetic apparatus, including water deficit and waterlogging [26,27].

In view of the scarce information about the response of kiwifruit seedlings to water stress and the need for sustainability in nursery production, the objective of this experiment was (i) to evaluate the growth and physiological responses of kiwifruit seedlings to different levels of SMC, and (ii) to define the optimum SMC for soilless cultivation of kiwifruit seedlings based on comprehensive analyses of root and shoot growth, antioxidant defense, photosynthesis, and chlorophyll fluorescence parameters. The results could provide a practical reference for high-quality kiwifruit seedling production in soilless cultivation system.

## 2. Materials and Methods

### 2.1. Plant Materials and Culture Conditions

Six-month-old tissue-cultured kiwifruit seedlings (*A. valvata* Dunn) were used, and the experiment was carried out in a greenhouse at the Zunjiu Kiwi orchard of Sichuan Huasheng Agriculture Co., Ltd. in Mianzhu, Sichuan, China (104°7′ E, 31°23′ N). The mean temperature was 29.18 °C, the relative humidity was 66.28%, and the light source was natural sunlight. Cylindrical plastic containers with a diameter of 30 cm and a height of 30 cm were filled with a mixture of coconut coir (Global Substrates, Inc., Sri Lanka, India), peat (Jiffy, Inc., Pärnu, Estonia), and perlite (Yunnan Hongchou Agricultural Technology Co., Ltd., Kunming, China) in a 1:2:2 volume ratio. The physicochemical properties of the mixture were as follows: pH 5.42, electrical conductivity (EC) 0.72 ms·cm$^{-1}$, bulk density 0.15 g·cm$^{-3}$, total porosity 69.95%, aeration porosity 19.97%, water-holding porosity 49.99%, and void ratio 0.40. The mixture was composed of 0.51% total N, 47.12% organic matter, 1049.54 mg·kg$^{-1}$ alkali-hydrolyzed nitrogen, 239.61 mg·kg$^{-1}$ available phosphorus, and 2194.16 mg·kg$^{-1}$ available potassium.

### 2.2. Substrate Moisture Content, Irrigation, and Plant Growth

The SMC was calculated as [weight of the water-saturated substrate − dry weight of substrate]/dry weight of substrate × 100%. Uniform kiwifruit seedlings (13.03 g per plant) were transplanted to the containers with one seedling per container on 20 May 2022. All substrate in the containers was maintained at saturation capacity for one week for root establishment, after which the plants were divided into 20%, 40%, 60%, 80%, 100%, and 120% SMC treatments with five replicate containers per treatment. Substrate moisture was measured based on the difference between the predetermined SMC value and the weight of the container every day at 17:00 [28]. Irrigation was started when the SMC dropped below the SMC of a treatment: the 100% and 120% SMC treatments were initiated on 27 May, 80% SMC on 13 June, 60% SMC on 22 June, 40% SMC on 1 July, and 20% SMC on 29 July. The volume of water and the number of irrigation events needed to maintain the SMC level were recorded. On 28 August, the plants were harvested, and the dry weights of roots and shoots were determined by oven-drying to constant weight at 70 °C.

### 2.3. Measurement of Chlorophyll Content and Photosynthetic Parameters

To determine the leaf Chl content, 0.2 g of fresh tissue was extracted in 95% (*v*/*v*) ethanol for 24 h in the dark. The absorbance values of the solution were measured at 649 and 665 nm using an ultraviolet spectrophotometer (Shimadzu UV-1800, Japan) [29]. Photosynthetic parameters in leaves were measured using a LI-6800 portable photosynthetic analyzer (Li-Cor, Inc., Lincoln, NE, USA). Measurements were taken on sunny days between 9:00 and 12:00. Five consecutive leaves in the middle of the stem (the 25th leaf from the root) were selected from each container for measurements of $P_n$, $g_s$, $T_r$, and intercellular $CO_2$ concentration ($C_i$). A fully expanded leaf was placed in the leaf chamber (25 °C, 400 μL L$^{-1}$ $CO_2$, and 1000 μmol photon m$^{-2}$ photosynthetic photon flux). The instantaneous carboxylation efficiency (ICE) and water use efficiency (WUE) were calculated using the following respective formulas: ICE = $P_n/C_i$ and WUE = $P_n/T_r$. The same fully expanded leaves were subjected to darkness for 20 min at 12:00, and the dark-adapted leaves were used to determine the Chl fluorescence parameters, including maximal quantum yield of PSII photochemistry ($F_v/F_m$), variable fluorescence out of minimal fluorescence yield under dark-adapted state ($F_v/F_0$), photochemical quenching ($q_p$), non-photochemical quenching (NPQ), actual quantum yield ($Y_{(II)}$), and photosynthetic electron transfer rate (ETR) using a portable Chl fluorescence meter (MINI-PAM-II, Walz, Germany).

### 2.4. Measurement of Leaf Relative Water Content and Cell Membrane Stability

For measurement of leaf relative water content (RWC), the fresh weight (FW) of a leaf sample was recorded, the leaves were immersed in deionized water for 24 h, and the saturated weight (SW) was measured. Dry weight (DW) was subsequently determined after over-drying at 70 °C for 48 h to constant weight. RWC was calculated using the equation of Barrs and Weatherley [30]: RWC (%) = (FW − DW)/(SW − DW) × 100. For determination of electrolyte leakage (EL), 0.2 g of fresh leaves was rinsed with distilled water and then immersed in deionized water for 24 h, and the initial conductivity (*Ci*) was measured using a conductivity meter (YSI Model 32, Yellow Springs, OH, USA). Leaf samples were boiled at 100 °C for 30 min, and the terminal conductance was measured as maximum conductivity (*Cm*) after being cooled to room temperature. EL was calculated as the percentage of *Ci* to *Cm* [31].

### 2.5. Measurement of Antioxidant Enzyme Activity and Lipid Peroxidation

To analyze the malondialdehyde (MDA) content and antioxidant enzyme activities, including superoxide dismutase (SOD), peroxidase (POD), and catalase (CAT), 0.2 g of fresh leaf samples was ground in 4 mL of phosphoric acid buffer (50 mM, pH 7.6), and the supernatant was collected after centrifugation at 4 °C for 30 min at 12,000 × *g*. MDA content was measured by incubating 0.5 mL of supernatant with 1 mL of 20% (*w*/*v*) trichloroacetic acid (TCA) containing 0.5% (*w*/*v*) thiobarbituric acid (TBA) at 95 °C for 15 min, centrifuging

at $8000 \times g$ for 10 min, and measuring the absorbance values of the supernatant at 532 and 600 nm using a spectrophotometer (Shimadzu UV-1800, Japan) [32]. SOD activity was measured by adding 50 µL of supernatant to a 1.5 mL reaction solution containing 195 mM methionine, 3 µM EDTA, 1.125 mM NBT, 60 µM riboflavin, and 50 mM PBS buffer (pH 7.8), and measuring the absorbance value at 560 nm [33]. POD and CAT activities were determined using the method of Chance and Maehly [34], and the absorbance values of the reaction solution were measured at 470 and 240 nm, respectively.

### 2.6. Date Analysis and Statistics

A completely randomized design was implemented for the experiment with five replicates per treatment. Differences were tested using one-way analysis of variance (ANOVA) and Tukey's multiple comparisons test with SPSS Statistics 26 (IBM, Armonk, NY, USA). Differences between treatments were considered statistically significant at $p < 0.05$.

## 3. Results

### 3.1. Total Irrigation Volume and Irrigation Frequency

The irrigation frequency increased with increasing SMC up to 100% SMC, and the total irrigation volume increased up to 120% SMC (Figure 1). The irrigation volume increased linearly from 20% to 100% SMC ($R^2 = 0.983$; $p = 0.001$; range 4.94 to 62.91 L), whereas at 120% SMC, the irrigation volume was 2.33 times higher than that at 100% SMC. During the experimental period, the irrigation frequency or total amount of water applied for each treatment was 31, 59, 68, 77, 94, and 94 times or 4.94, 25.18, 38.19, 49.31, 62.91, and 146.86 L per container at 20%, 40%, 60%, 80%, 100%, and 120% SMC, respectively.

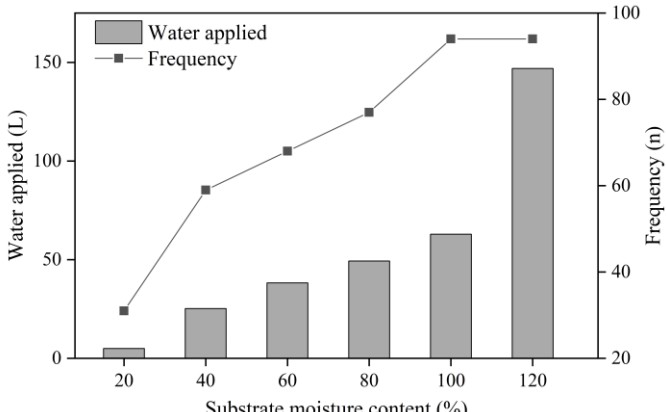

**Figure 1.** Total amount of water applied and irrigation frequency of kiwifruit seedlings at substrate moisture contents (SMCs) from 20% to 120% SMC.

### 3.2. Plant Growth Parameters

Shoot and root dry weights increased gradually with the increase in SMC from 20% to 100% and were lower at 120% SMC than at 100% SMC ($p < 0.05$) (Figure 2A,B). Compared to that at 100% SMC, shoot dry weight was reduced by 91.04%, 59.95%, 21.00%, 11.66%, and 45.44%, and root dry weight was reduced by 91.68%, 56.68%, 44.40%, 24.82%, and 12.15% at 20%, 40%, 60%, 80%, and 120% SMC, respectively. The root systems were clearly different at different SMCs (Figure 2C).

### 3.3. Cell Membrane Stability and Antioxidant Enzyme Activities

EL and MDA content were the lowest at 80% and 100% SMC. A 19.69%, 49.21%, 70.08%, 71.09%, or 57.52% decline in EL and 23.08%, 56.52%, 84.07%, 82.58%, or 50.82% decrease in MDA content were observed at 40%, 60%, 80%, 100%, and 120% SMC compared to that at 20% SMC, respectively (Figure 3). SOD, POD, and CAT activities were maintained at the highest levels at 80% and 100% SMC and declined with a gradual decrease in SMC ($p < 0.05$). The 120% SMC treatment also resulted in significant decreases in SOD, POD, and

CAT activities compared to those at 80% and 100% SMC ($p < 0.05$) (Figure 4). The SOD activity at 100% SMC was 23.24%, 14.17%, 9.53%, 2.27%, and 15.63% higher than that at 20%, 40%, 60%, 80%, and 120% SMC, respectively. Changes in POD and CAT activities were similar to those in SOD activity at different SMCs.

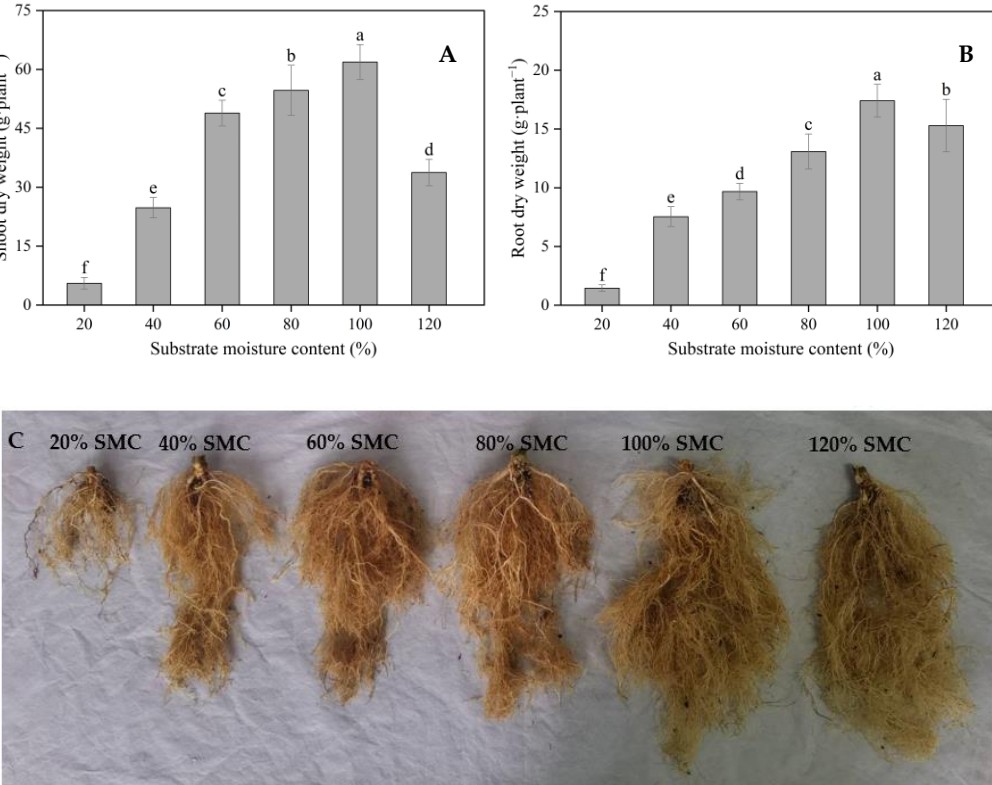

**Figure 2.** The impact of different SMCs on the shoot dry weight, root dry weight, and root morphology. Shoot and root dry weight (**A**,**B**) and morphology of the roots (**C**) of kiwifruit seedlings at SMCs from 20% to 120% SMC. Data are mean ± standard error (*n* = 5). Different letters above columns indicate statistically significant differences in the Tukey's multiple comparisons test ($p < 0.05$).

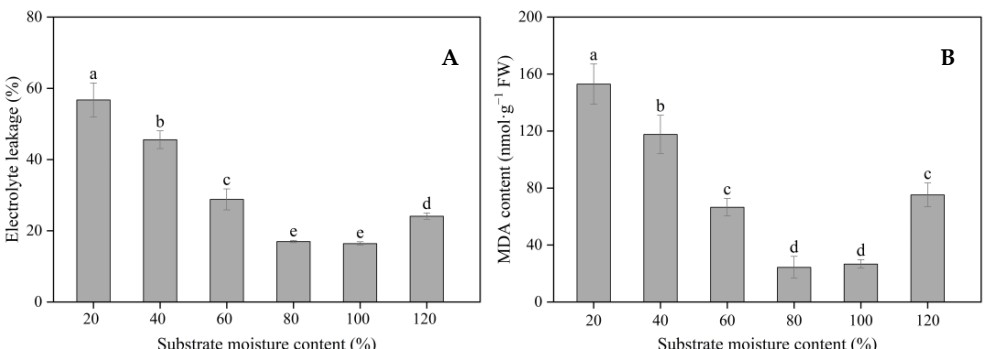

**Figure 3.** The impact of different SMCs on leaf EL and MDA content. EL (**A**) and MDA content (**B**) in kiwifruit seedlings at SMCs from 20% to 120% SMC. Data are mean ± standard error (*n* = 5). Different letters above columns indicate statistically significant differences in the Tukey's multiple comparisons test ($p < 0.05$).

### 3.4. Leaf Water Status and Chlorophyll Content

Leaf RWC increased with the increase in SMC from 20% to 80% and was the highest at 80%, 100%, and 120% SMC ($p < 0.05$) (Figure 5). Total Chl, Chl a, and Chl b contents increased with the increase in SMC from 20% to 80% and were lower at 120% SMC than at 80% and 100% SMC ($p < 0.05$) (Figure 6). Chl (a + b) content was 20.31%, 86.16%, 119.96%,

128.55%, and 65.06% higher, Chl a content was 20.21%, 120.97%, 156.89%, 156.98%, and 97.32% higher, and Chl b content was 9.10%, 48.19%, 74.01%, 80.55%, and 29.88% higher at 40%, 60%, 80%, 100%, and 120% SMC than at 20% SMC, respectively.

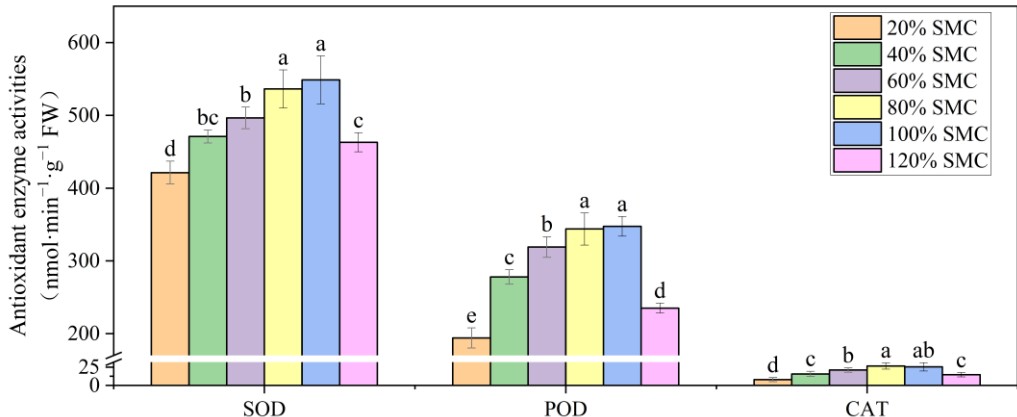

**Figure 4.** The impact of different SMCs on leaf SOD, POD, and CAT activities in kiwifruit seedlings at SMCs from 20% to 120% SMC. Data are mean ± standard error ($n$ = 5). Different letters above columns indicate statistically significant differences in the Tukey's multiple comparisons test ($p < 0.05$).

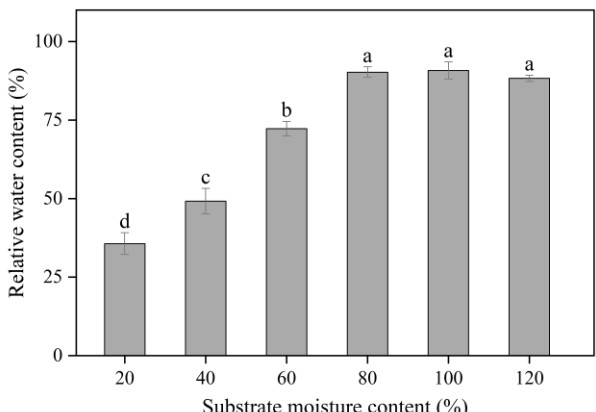

**Figure 5.** The impact of different SMCs on leaf RWC in kiwifruit seedlings at SMCs from 20% to 120% SMC. Data are mean ± standard error ($n$ = 5). Different letters above columns indicate statistically significant differences in the Tukey's multiple comparisons test ($p < 0.05$).

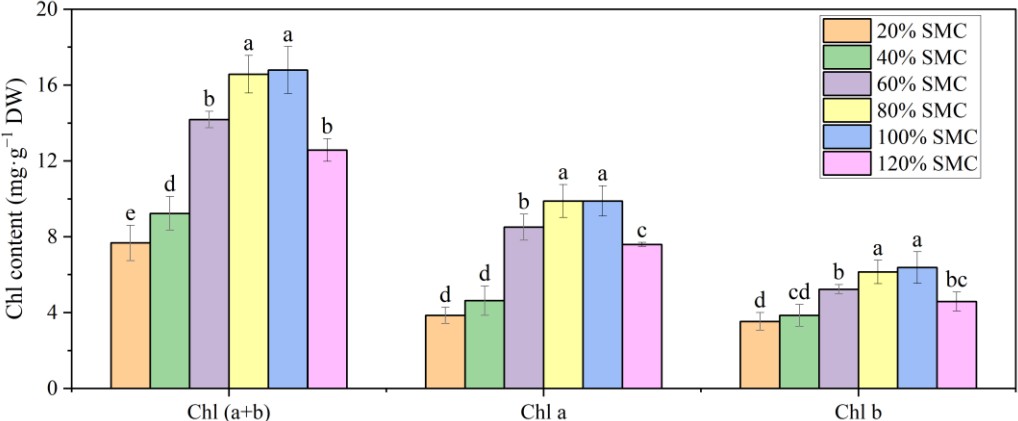

**Figure 6.** The impact of different SMCs on leaf (Chl (a + b), Chl a, and Chl b contents in kiwifruit seedlings at SMCs from 20% to 120% SMC. Data are mean ± standard error ($n$ = 5). Different letters above columns indicate statistically significant differences in the Tukey's multiple comparisons test ($p < 0.05$).

### 3.5. Photosynthetic Parameters

The photosynthetic parameters of the seedling leaves were significantly affected by different SMC treatments. $P_n$ increased with the increase in SMC from 20% to 80% and was lower at 120% SMC than at 80% and 100% SMC ($p < 0.05$) (Figure 7A). $g_s$ and ICE increased with the increase in SMC from 20% to 100% ($p < 0.05$) (Figure 7B,E). $C_i$ was the lowest at 80% and 100% SMC ($p < 0.05$) (Figure 7C). $T_r$ increased with the increase in SMC from 20% to 100% ($p < 0.05$) and was at the same level at 100% and 120% SMC (Figure 7D). Leaf WUE was the highest at 60% and 80% SMC and the lowest at 20% SMC ($p < 0.05$); it was reduced by 64.27%, 53.87%, 0.31%, 21.61%, and 55.10% at 20%, 40%, 80%, 100%, and 120% SMC compared to that at 60% SMC, respectively (Figure 7F).

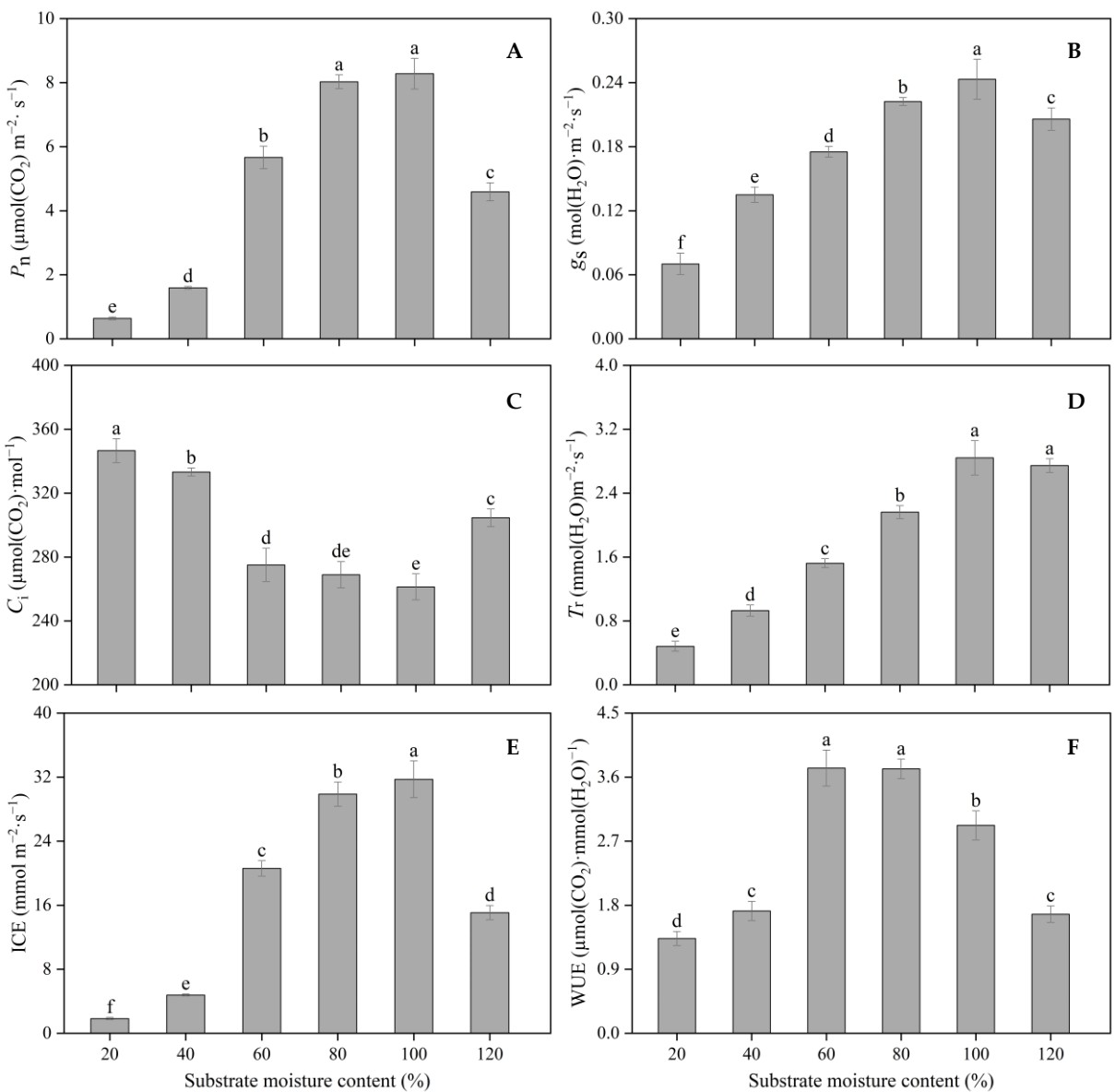

**Figure 7.** The impact of different SMCs on leaf photosynthetic parameters: $P_n$ (**A**), $g_s$ (**B**), $C_i$ (**C**), $T_r$ (**D**), ICE (**E**), and WUE (**F**) in kiwifruit seedlings at SMCs from 20% to 120% SMC. Data are mean ± standard error ($n = 25$). Different letters above columns indicate statistically significant differences in the Tukey's multiple comparisons test ($p < 0.05$).

$F_v/F_m$, ETR, and $Y_{(II)}$ were higher at 60% to 100% SMC than at other SMC treatments ($p < 0.05$). $F_v/F_0$ increased with the increase in SMC from 20% to 80% and was lower at 120% SMC than at 80% and 100% SMC ($p < 0.05$). $q_P$ was higher at 80% SMC than at 20%,

40%, and 120% SMC ($p < 0.05$). NPQ was the highest at 20% and 40% SMC ($p < 0.05$) and decreased gradually from 20% to 100% SMC (Table 1).

**Table 1.** The impact of different SMCs on leaf chlorophyll fluorescence parameters of kiwifruit seedlings at SMCs from 20% to 120% SMC.

| SMC (%) | $F_v/F_m$ | $F_v/F_0$ | ETR | $q_p$ | $Y_{(II)}$ | NPQ |
|---|---|---|---|---|---|---|
| 20 | $0.68 \pm 0.05$ [c] | $2.16 \pm 0.37$ [d] | $20.91 \pm 1.95$ [c] | $0.81 \pm 0.02$ [d] | $0.47 \pm 0.03$ [c] | $0.89 \pm 0.16$ [a] |
| 40 | $0.73 \pm 0.01$ [b] | $2.68 \pm 0.18$ [c] | $21.49 \pm 0.42$ [bc] | $0.86 \pm 0.02$ [bc] | $0.49 \pm 0.01$ [c] | $0.88 \pm 0.07$ [a] |
| 60 | $0.77 \pm 0.01$ [a] | $3.33 \pm 0.25$ [b] | $24.33 \pm 0.87$ [a] | $0.88 \pm 0.01$ [ab] | $0.55 \pm 0.02$ [a] | $0.69 \pm 0.07$ [b] |
| 80 | $0.79 \pm 0.01$ [a] | $3.75 \pm 0.22$ [a] | $25.25 \pm 0.33$ [a] | $0.90 \pm 0.02$ [a] | $0.57 \pm 0.01$ [a] | $0.64 \pm 0.07$ [b] |
| 100 | $0.79 \pm 0.01$ [a] | $3.77 \pm 0.37$ [a] | $25.05 \pm 0.27$ [a] | $0.88 \pm 0.01$ [ab] | $0.57 \pm 0.01$ [a] | $0.59 \pm 0.04$ [b] |
| 120 | $0.72 \pm 0.03$ [b] | $2.63 \pm 0.38$ [c] | $22.77 \pm 1.11$ [b] | $0.84 \pm 0.03$ [c] | $0.52 \pm 0.03$ [b] | $0.67 \pm 0.08$ [b] |

$F_v/F_m$, maximal quantum yield of PSII photochemistry; $F_v/F_0$, variable fluorescence out of minimal fluorescence yield under dark-adapted state; ETR, photosynthetic electron transfer rate; $q_p$, photochemical quenching; $Y_{(II)}$, actual quantum yield; NPQ, non-photochemical quenching. Data are mean $\pm$ standard error ($n = 25$). Different letters in a column indicate a statistically significant difference in Tukey's multiple comparisons test ($p < 0.05$).

## 4. Discussion

Optimum SMC can decrease cultivation costs and improve the quality of seedlings in soilless cultivation. Similar to studies on the irrigation of petunias (*Petunia × hybrida*) in which no irrigation water leached out of containers [35], the total irrigation volume increased linearly with SMC up to 100% SMC. In agreement with the results of Burnett and van Iersel [36], who reported that substantially more water was needed as a result of leaching due to an SMC close to or higher than the water-holding capacity of the substrate, the total irrigation volume was 2.33 times higher at 120% than at 100% SMC. Thus, exceeding the SMC resulted in earlier irrigation implementation with higher frequency and a larger amount of water consumed (Figure 1).

When plants are subjected to water stress, such as water deficit or waterlogging, their physiological and metabolic processes are disrupted leading to growth retardation [37,38]. Drought stress restricts the transport of water and nutrients from roots to leaves [39,40]. Drought-induced dehydration stimulates the biosynthesis of phytohormones such as abscisic acid (ABA) in the root system, which can be transferred to leaf guard cells via xylem transport to regulate stomatal closure, thus resulting in a decrease in transpiration and water loss from leaves [18,41,42]. Excessive water in soils leads to an oxygen-deficient environment around the roots, which suppresses root aerobic respiration and causes an energy shortage [20]. In addition, waterlogging also induces the accumulation of ROS and toxic metabolites such as ethanol and lactic acid, which inhibit plant growth and may even result in death of the root system [18,43]. In our study, the lower shoot and root biomass at up to 60% SMC as well as at 120% SMC indicated that the inhibition of shoot and root growth could be caused by drought or waterlogging stress, suggesting that 80% and 100% SMC were appropriate for soilless cultivation of kiwifruit seedlings (Figure 2A,B). Interestingly, our findings indicated that the waterlogging-induced negative effect on root growth was less than that induced by water deficit (Figure 2C). A previous study by Li et al. [44] also found that *A. valvata* vines could endure long-term waterlogging stress by sacrificing most of the main root system.

Under normal water conditions, the production and elimination of ROS in plants are in dynamic equilibrium. Water stress disturbs homeostasis by limiting the photosynthetic process, resulting in an increase in ROS production [45,46]. It is well documented that drought and waterlogging stress induce the activation of the antioxidant defense system to avoid oxidative damage in plants. SOD, CAT, and POD are the primary antioxidant enzymes involved in the antioxidant defense system, in which SOD catalyzes the conversion of $O^{-2}$ into $H_2O_2$, and POD and CAT catalyze the reduction of $H_2O_2$ to $H_2O$ and $O_2$ [47]. In general, plants increase the activities of various antioxidant enzymes to cope with water stress [48–50]. In our study, higher MDA content and EL levels at up to 60% SMC and at

120% SMC suggested that both water deficit and waterlogging stress accelerated membrane lipid peroxidation and damaged cell membrane stability (Figure 3), which agreed with previous findings indicating that prolonged drought or waterlogging stress disturbed antioxidant metabolism and ROS homeostasis [51–54]. Significantly lower phenolic content and antioxidant capacity per shoot in *Crepidiastrum denticulatum* were observed under drought and flooding stress [20]. In the present study, the reduced SOD, POD, and CAT activities in leaves of *A. valvata* Dunn when subjected to water deficit or waterlogging indicated that the antioxidant capacity of kiwifruit was weakened by water stress, and the overaccumulation of ROS could not be scavenged (Figure 4).

One of the major strategies employed by plants to respond to water stress is to reduce transpiration by closing stomata, which leads to a reduction in the leaf photosynthetic rate due to reduced gas interchange [21,55,56]. An et al. [23] reported that the net $CO_2$ assimilation rate, $g_s$, and $T_r$ in cymbidium (*Cymbidium* spp.) increased with the increase in substrate volumetric water content. Our data showed that $P_n$ and $g_s$ of *A. valvata* Dunn leaves decreased at up to 60% and at 120% SMC. However, $C_i$ was not limited by $g_s$ but increased markedly in response to water deficit (20% and 40% SMC) and waterlogging (120% SMC) stress (Figure 7), indicating that the decrease in $P_n$ was not related to the reduction in $g_s$. This was consistent with previous studies showing that the negative effect of severe drought or waterlogging stress on photosynthesis was mainly caused by nonstomatal factors [57,58]. Nonstomatal limitations such as stress-induced Chl degradation, damage to the photosynthetic apparatus, and a reduction in photosynthetic enzymatic activity contribute to the decline in photosynthetic $CO_2$ assimilation and $P_n$ [18,59]. ICE has been widely used to estimate the activity of RuBisCO, which is a key enzyme involved in $CO_2$ assimilation [60]. Silva et al. [56] found that water stress significantly decreased ICE of sugarcane (*Saccharum* spp.). These results further confirmed that the decreases in Chl content and photosynthetic enzyme activity due to long-term drought and waterlogging stress inhibit photosynthesis. In addition, nitrogen deficiency caused by leaching under waterlogging stress could further promote the decrease in photosynthetic capacity [59]. Plants tend to adapt to water deficit stress by increasing WUE [61]. In line with that, WUE of *A. valvata* Dunn was the highest at 60% and 80% SMC and the lowest at 120% SMC.

The health status of PSII and the photosynthetic electron transport chain can be evaluated by analyzing the induction curve of Chl fluorescence in plants, since Chl fluorescence reflects the intrinsic properties of the photosynthetic system [26,62]. Chl fluorescence kinetics have been widely used in studying water stress physiology [63,64]. Leaf $F_v/F_m$ indicates the maximum photochemical efficiency of PSII, which has been proven to be a good physiological reference for diagnosing the integrity of plant photosynthetic systems under stressful conditions [26,56,65]. Similar to previous studies in which water stress decreased photochemical efficiency [56,66], leaf $F_v/F_m$ and $F_v/F_0$ were lower at 20%, 40%, and 120% SMC than under normal conditions (80% and 100% SMC). In addition, $q_p$ provides insight into the proportion of open reaction centers in PSII, and its increase is positively correlated with electron transport activity [26]. Zhang et al. [21] found that $q_p$ decreased markedly in response to drought and waterlogging stress, accompanied by reductions in ETR and $Y_{(II)}$. Bañón et al. [22] also reported that ETR of *Hebe andersonii* decreased with the decrease in substrate volumetric water content, while NPQ was the highest under more severe water-deficit conditions. Higher values of NQP were observed under water stress as a result of energy dissipation [26,67]. Similar results were demonstrated in the current study, indicating that both water deficit and waterlogging stress could be harmful to the photochemical system of *A. valvata* Dunn seedlings.

## 5. Conclusions

Kiwifruit seedlings showed significant differences in growth and physiological parameters under different SMCs. The main results were as follows: (i) shoot and root dry weights of seedings increased with the increase in SMC from 20% to 100% but were lower at 120% SMC than at 100% SMC; (ii) seedlings at lower SMC (20%, 40%, and 60% SMC)

as well as at 120% SMC exhibited significant lower activities of SOD, POD, and CAT and higher EL and MDA accumulation than seedlings at 80% and 100% SMC; (iii) Higher $P_n$, $g_s$, ICE, and chlorophyll fluorescence parameters, including $F_v/F_m$, $F_v/F_0$, ETR, and $Y_{(II)}$, were observed at 80% and 100% SMC. Combined with the growth and physiological changes, the biomass could be used as a main evaluation parameter for selection of the optimal substrate moisture content for the growth of kiwifruit seedlings since most of physiological parameters were not significantly different between 80% and 100% SMC, but seedling achieved the highest biomass at 100% SMC. Therefore, 100% SMC provided the optimal water supply for photosynthetic efficiency and dry matter accumulation of shoots and roots. The results are expected to be useful for the mass production of high-quality kiwifruit seedlings in greenhouse or nursery containers, with the potential to save water.

**Author Contributions:** Conceptualization, Y.-X.C. and K.-W.X.; investigation, D.-G.C., H.-Y.Y. and R.Y.; data curation, D.-G.C., D.-D.P. and R.Y.; formal analysis, D.-G.C. and D.-D.P.; supervision, Y.-X.C., D.-D.P. and H.-P.L.; project administration, Y.-X.C. and K.-W.X.; funding acquisition, Y.-X.C. and H.-P.L.; writing—original draft, D.-D.P. and D.-G.C.; writing—review and editing, P.P. and K.-W.X. All authors have read and agreed to the published version of the manuscript.

**Funding:** This research was funded by the Sichuan Key Research and Development Project (2021YFN 0026) and Sichuan Science and Technology Project (2022ZHXC0007, 2023NSFSC0138).

**Data Availability Statement:** Data sharing is not applicable.

**Conflicts of Interest:** The authors declare no conflict of interest.

## Abbreviations

SMC, substrate moisture content; $P_n$, net photosynthetic rate; $g_s$, stomatal conductance; $T_r$, transpiration rate; $C_i$, intercellular $CO_2$ concentration; PSII, photosystem II; Chl, chlorophyll; WUE, water use efficiency; ICE, instantaneous carboxylation efficiency; $F_v/F_m$, maximal quantum yield of PSII photochemistry; $F_v/F_0$, variable fluorescence out of minimal fluorescence yield under dark-adapted state; $q_p$, photochemical quenching; NPQ, non-photochemical quenching; $Y_{(II)}$, actual quantum yield; ETR; photosynthetic electron transfer rate; RWC, relative water content; EL, electrolyte leakage; MDA, malondialdehyde; SOD, superoxide dismutase; POD, peroxidase; CAT, catalase; ROS, reactive oxygen species.

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
