# Peer review of "Optimal Substrate Moisture Content for Kiwifruit (Actinidia valvata Dunn) Seedling Growth Based on Analyses of Biomass, Antioxidant Defense, and Photosynthetic Response"

_agronomy, doi:10.3390/agronomy13071858_

Round 1

Reviewer 1 Report

Reviewing report regarding the research article entitled ‘’Growth, Antioxidant, and Photosynthetic Responses of Kiwifruit (Actinidia valvata Dunn) Seedlings to Different Substrate Moisture Content’’.

The research work is interesting, but may I suggest you correct the following:

Regarding the title, please change it to make it more attractive to readers.

Lines from 70-74: please rewrite the aim of your research.

Introduction: Please add in the introduction paragraph related to Reactive Oxygen Species.

Please add more recent citations related to your research work.

Material and Methods: OK

Results: Please add the title to your figures because it seems that you describe your figures without adding the title.

For Example,

Figure 2. The impact of substrate moisture contents (SMC) at different percentages on the shoot dry weight, dry weight, and root morphology. Shoot and root dry weights (A and B) and morphology of the roots (C) of kiwifruit seedlings at substrate moisture contents (SMC) from 20% to 120% SMC. Data are mean ± SE (n = 5). Different letters above columns indicate a statistically significant difference in the Tukey’s multiple comparison tests (p < 0.05).

Table 1: please make the significant litter superscript.

Discussion: OK

Conclusion: OK

Quality of English language is ok 

Author Response

Point 1: Regarding the title, please change it to make it more attractive to readers.

Response 1: Thank you for your advice. The title has been revised to “Optimal Substrate Moisture Content for Kiwifruit (Actinidia valvata Dunn) Seedlings growth based on analyses of biomass, Antioxidant, and Photosynthetic Response” according to your suggestion (Line 4 - 6).

Point 2: Lines from 70-74: please rewrite the aim of your research.

Response 2: Thanks. The aim of the research has been rewrote according to to your suggestions in revised manuscript (Line 85 - 92).

Point 3: Introduction: Please add in the introduction paragraph related to Reactive Oxygen Species.

Response 3: Thank you very much for your professional review. We have added Reactive Oxygen Species in Introduction (Line 78 - 81).

Point 4: Please add more recent citations related to your research work.

Response 4: Thanks. We have added more recent citations related to our research work in revised manuscript (see below the references).

[22] Banon, D.; Lorente, B.; Banon, S.; Ortuno, M. F.; Sanchez-Blanco, M. J.; Alarcon, J. J., Control of Substrate Water Availability Using Soil Sensors and Effects of Water Deficit on the Morphology and Physiology of Potted Hebe andersonii. Agronomy-Basel 2022, 12, (1).

[23] An, S. K.; Lee, H. B.; Kim, J.; Kim, K. S., Efficient Water Management for Cymbidium Grown in Coir Dust Using a Soil Moisture Sensor-Based Automated Irrigation System. Agronomy-Basel 2021, 11, (1).

[25] Jia, L.; Qin, X.; Lyu, D.; Qin, S.; Zhang, P., ROS production and scavenging in three cherry rootstocks under short-term waterlogging conditions. Scientia Horticulturae 2019, 257.

Point 5: Results: Please add the title to your figures because it seems that you describe your figures without adding the title.

For Example,

Figure 2. The impact of substrate moisture contents (SMC) at different percentages on the shoot dry weight, dry weight, and root morphology. Shoot and root dry weights (A and B) and morphology of the roots (C) of kiwifruit seedlings at substrate moisture contents (SMC) from 20% to 120% SMC. Data are mean ± SE (n = 5). Different letters above columns indicate a statistically significant difference in the Tukey’s multiple comparison tests (p < 0.05).

Response 5: Thank you very much for your professional and careful review. We have added the title to the figures and table throughout the paper according to your suggestion.

Point 6: Table 1: please make the significant litter superscript.

Response 6: Thanks. We have made the the significant litter superscript in the Table 1 according to your suggestion.

Reviewer 2 Report

Manuscript ID: Agronomy-2484605

“Growth, Antioxidant, and Photosynthetic Responses of Kiwifruit (Actinidia valvata Dunn) Seedlings to Different Substrate Moisture Content

General comments

It is known that plants are susceptible to environmental factors such as drought or waterlogging. In this work, the authors study the impact on the use of Optimum substrate moisture content to estimate its water requirements in a greenhouse study. For this purpose, they carry out different studies based on the growth, antioxidant defense, and photosynthetic parameters of kiwifruit (Actinidia valvata Dunn) seedlings at six levels of SMC (20%, 40%, 60%, 80%, 100%, and 120%). Results showed that 100% SMC provided an optimal water supply for the photosynthetic efficiency and dry matter accumulation of shoots and roots. The results are expected to be useful for the mass production of high quality kiwifruit seedlings in greenhouse or nursery containers with potential to save water.

Comments:

1)       In the abstract, Line 17: kiwifruit (Actinidia valvata Dunn), the authors indicate a particular species, but in line 33 consider family (Actinidia sp.). Why does it make that difference?

2)       Line 112: Tr, and intercellular CO2 concentration (Ci),.

remove the comma

3)       Line 114: The instantaneous carboxylation efficiency (CUE),

Why you use CUE as abbreviation for carboxylation efficiency? It's confusing, it would be better CE

4)       Figure 2 b, the percentage for column f is correct? It seems to be smaller in the graph than indicated in the text.

5)       Homogenize the use of abbreviations, in some graphs it uses them and in others it does not. More than once in the manuscript the meaning of some abbreviations is indicated (see text and figure legends), put only the first time it appears.

6)       Figure 4: the column for 60% of SOD is not the correct color; it should be purple, not light blue. Idem figure 6 in Chl (a+b)

7)       Lines 242 to 246: please introduce before the table 1

Author Response

Point 1: In the abstract, Line 17: kiwifruit (Actinidia valvata Dunn), the authors indicate a particular species, but in line 33 consider family (Actinidia sp.). Why does it make that difference?

Response 1: Yes, “Actinidia valvata Dunn”is one of the species of Actinidia sp., which was used as the experimental material of this study. To aviod confusion, the “parameters of kiwifruit (Actinidia valvata Dunn) seedlings” in the abstract has been replaced by “parameters of seedlings of Actinidia valvata Dunn” in revised manuscript (Line 20).

Point 2: Line 112: Tr, and intercellular CO2 concentration (Ci),. remove the comma

Response 2: The comma after “and intercellular CO2 concentration (Ci)” has been removed (Line 130).

Point 3: Line 114: The instantaneous carboxylation efficiency (CUE),

Why you use CUE as abbreviation for carboxylation efficiency? It's confusing, it would be better CE

Response 3: Thank you very much for your careful review. The abbreviation of “instantaneous carboxylation efficiency” is more appropriate as “ICE”, and we have changed the “CUE” to “ICE” throughout the paper.

Point 4: Figure 2 b, the percentage for column f is correct? It seems to be smaller in the graph than indicated in the text.

Response 4: Thanks. We have double checked the data, and it is correct.

Point 5: Homogenize the use of abbreviations, in some graphs it uses them and in others it does not. More than once in the manuscript the meaning of some abbreviations is indicated (see text and figure legends), put only the first time it appears.

Response 5: Thank you for your advice. We have been homogenized the abbreviations throughout the paper.

Point 6: Figure 4: the column for 60% of SOD is not the correct color; it should be purple, not light blue. Idem figure 6 in Chl (a+b)

Response 6: Thank you very much for your careful review. The color of column for 60% of SOD and Chl (a+b) in Figure 4 and Figure 6 have been corrected.

Point 7: Lines 242 to 246: please introduce before the table 1

Response 7: Thanks. We have put the introduce before the table 1 according to your suggestion (Line 257 - 261). 

Reviewer 3 Report

The presented article is devoted to the selection of the optimal moisture conditions for kiwifruit multiplication. This topic is really important because different plants require different conditions for the development. At the same time it is important to optimize the process to reduce the expenses needed for this. To estimate water requirements of the seedlings in a greenhouse authors analyzed the growth, antioxidant defense, and photosynthetic parameters of kiwifruit seedlings at six levels of substrate moisture content (20%, 40%, 60%, 80%, 100%, and 120%). It is important that the experiment was complex and many parameters were evaluated in several conditions. It was found that shoot and root dry matter accumulations increased gradually with the increase in substrate moisture content from 20% to 100%, and were lower at 120% of substrate moisture content than at 100% of it. Among other parameters, electrolyte leakage and malondialdehyde contents were the lowest at 80% and 100% substrate moisture content. 100% SMC provided an optimal water supply for the photosynthetic efficiency and dry matter accumulation of shoots and roots.

It is very good, that authors described in details all used in the experiment methods and listed many articles, connected with their investigation.

At the same time I have some remarks.

1. Article contains many abbreviations. it would be useful to add the list of abbreviations.

2. Chapter “Conclusion” is absent.

Author Response

Point 1: Article contains many abbreviations. it would be useful to add the list of abbreviations.

Response 1: Thank you very much for your professional and careful review. The list of abbreviations has been added in revised manuscript (Line 36 - 43).

Point 2: Chapter “Conclusion” is absent.

Response 2: The last paragraph of the Discussion is the conclusion, and we have rewritten the section and the title of “Conclusions” has been added before it in revised manuscript (Line 358 -374).

Reviewer 4 Report

The article entitled,, Growth, Antioxidant, and Photosynthetic Responses of Kiwifruit (Actinidia valvata Dunn) Seedlings to Different Substrate Moisture Content' raises important issues concerning Kiwifuit cultivation and contains interesting analyses. This work extends the current state of knowledge about this plant. Unfortunately, the article has several weaknesses that need to be corrected to make it suitable for publication.

The final results for the individual parameters should be followed by a thorough discussion of the results obtained. The results should be compared with those previously published in the literature. This is currently lacking in the paper.

The discussion chapter needs to be rewritten. The authors have not paid sufficient attention to the correlations obtained. They have not explained which factors and mechanisms influence the final result. It is also important to show a potential direction for future research.

The paper lacks a conclusion chapter. Without it, it is unclear what added value the study provided to science. Please pore over the main results as a bullet point list. I believe the paper should not be published without this chapter.

Line 16: please do not use the ,,we" form but the impersonal form, the same in line 71.

Moderate editing of English language required

Author Response

Point 1: The final results for the individual parameters should be followed by a thorough discussion of the results obtained. The results should be compared with those previously published in the literature. This is currently lacking in the paper.

Response 1: Thank you for your suggestion. We have added relevant discussions which compared our current findings with those previously published literature in the section of Discussion (Line 310-312, 319-321, and 351-353).

Point 2: The discussion chapter needs to be rewritten. The authors have not paid sufficient attention to the correlations obtained. They have not explained which factors and mechanisms influence the final result. It is also important to show a potential direction for future research.

Response 2: Thanks. We have revised the discussion section and added some explanations in Conclusion section according to your suggestion: biomass could be used as a main evaluation parameter for selection of optimal substrate moisture content for growth of kiwifruit seedling, since most of physiological parameters were not significantly different between 80% and 100% SMC, but seedling achieved the highest biomass at 100%. We have pointed out a potential direction for future research in the section of Conclusions: 100% SMC provided an optimal water supply for the photosynthetic efficiency and dry matter accumulations of shoots and roots. The results are expected to be useful for the mass production of high-quality kiwifruit seedlings in greenhouse or nursery containers with potential to save water.

Point 3: The paper lacks a conclusion chapter. Without it, it is unclear what added value the study provided to science. Please pore over the main results as a bullet point list. I believe the paper should not be published without this chapter.

Response 3: Thanks. The conclusion chapter has been added in revised manuscript and we also added a bullet point list to show the main results according to your suggestion (Line 358-374).

Point 4: Line 16: please do not use the “we" form but the impersonal form, the same in line 71.

Response 4: Thank you very much for your professional and careful review. We have deleted the “we” throughout the manuscript and revised these sentences according to your suggestions carefully (Line 16 and 85-92).